# Rhodium-catalysed direct hydroarylation of alkenes and alkynes with phosphines through phosphorous-assisted C−H activation

Dingyi Wang[1,2], Ben Dong[1,2], Yandong Wang[1], Jiasheng Qian[1], Jinjun Zhu[1], Yue Zhao[1] & Zhuangzhi Shi [1]

Biarylphosphines have been widely applied as ligands in various synthetic methods, especially in transition-metal-catalysed carbon-carbon and carbon-heteroatom bond cross-coupling reactions. Based on the outstanding properties of the parent scaffolds, a general method for in situ modification of the commercial tertiary phosphine ligands to access a series of ligands is in high demand. Here we show that a rhodium-catalysed system is introduced for the hydroarylation of alkenes and alkynes with tertiary phosphines through P(III)-chelation assisted C-H activation. A series of ligand libraries containing alkyl and alkenyl substituted groups with different steric and electronic properties are obtained in high yields. Furthermore, several experimental studies are performed to uncover the key mechanistic features of the linear-selective hydroarylation of alkenes and branch-selective hydroarylation of alkynes.

[1] State Key Laboratory of Coordination Chemistry, School of Chemistry and Chemical Engineering, Nanjing University, 210093 Nanjing, China. [2] These authors contributed equally: Dingyi Wang, Ben Dong. Correspondence and requests for materials should be addressed to Z.S. (email: shiz@nju.edu.cn)

Transition-metal-catalysed C−H activation assisted by chelating directing groups is a well-established strategy[1–10]. Of these processes, the hydroarylation of C−C double and triple bonds through C−H addition has received much attention recently because it represents a versatile and atom-economical method[11–14]. A remarkable feature of chelation-assisted hydroarylation resides in the ability to overcome the inert nature of C−H bonds and control site-selectivity in molecules with diverse C−H bonds. In 1993, Murai et al.[15] reported a significant, pioneering work on the Ru-catalysed hydroarylation of alkenes with aromatic ketones by carbonyl-directed C−H activation. Since then, the emergence of promising O-chelation assisted hydroarylations in various substrates such as N,N-dimethylcarbamoyl indoles[16], N,N-diethylbenzamides[17,18], anilides[19], enamides[20–23] and so on by rhodium and iridium catalysts has been reported (Fig. 1a)[24]. In the ensuing years, the catalytic hydroarylation reactions have focused on the development of N directing groups with different transition metal catalysts (Fig. 1b)[25–34]. Since 2010, Yoshikai and co-workers[35–40] have reported a series of cobalt-catalysed hydroarylation of alkenes and alkynes with 2-phenylpyridines and imines. In 2013, Wang and Chen uncovered the Mn(I)-catalysed aromatic C−H alkenylation of terminal alkynes with 2-phenylpyridines[41,42]. In 2018, Ackermann and co-workers[43] reported the enantioselective cobalt(III)-catalysed C-H alkylation of indole at C2 position with alkenes by an N-pyridine-type directing group. A very recent example by Yu, Sun and Lu involved a U-shaped nitrile template-directed, Rh(III)-catalysed meta-C−H alkenylation of arenes with disubstituted alkynes, wherein the chelation of cyano group was crucial for the meta-selective hydroarylation[44]. Despite these widely acknowledged advances, a variation to the original Murai protocol, which enables catalytic P-chelated, regioselective hydroarylation remains elusive.

P-chelated transition metal-olefin complexes[45] with an aromatic C−H bond have been studied from a stoichiometric point of view. In 2011, Schauer and Brookhart uncovered the migratory insertion of square-pyramidal d[6], 16-electron Ir(III), and Rh(III) olefin hydride complexes assisted by a (bis)chelated O-P(III) motif (Fig. 1c)[46]. Due to the high stability of pincer complexes[47,48], subsequent reductive elimination to build the C(sp[2])–C(sp[3]) bond

could not proceed. Furthermore, the catalytic variation is even more challenging. Recently, we[49] and Clark[50,51] have disclosed C–H arylation and borylation of biaryl phosphines, in which the inherent monodentate phosphine was used as a directing group (Fig. 1d). The development of the P[III]-chelation-assisted C–H activation in a catalytic process prompted us to consider whether these phosphines could directly undergo hydroarylation by transition metal catalysts[52–58]. Inspired by these previous results, herein, we report a catalytic transformation for the regioselective hydroarylation of commercially available tertiary phosphines with alkenes and alkynes through a rhodium-catalysed, P(III)-directed C–H alkylation and alkenylation (Fig. 1e). Such routes are particularly desirable because alkyl and alkenyl-substituted phosphine ligands[59–61] with different steric and electronic properties can be produced by an in situ modification strategy[62].

## Results

**Reaction design.** We initiated our study by investigating the reaction of phosphine **1a** with methyl acrylate (**2a**) in the presence of an array of rhodium and iridium catalysts (Table 1). As a result, we discovered that the use of 2.5 mol% [Rh(cod)Cl]$_2$ and 3.0 equivalents of NaHCO$_3$ in toluene at 140 °C for 3 h led to the formation of alkylation product **3aa** with a 75% yield and a small amount of disubstituted alkylation product **3aa′** (Table 1, entry 1). Other rhodium complexes such as [Rh(coe)$_2$Cl]$_2$ (Table 1, entry 2) and [Rh(CO)$_2$Cl]$_2$ (Table 1, entry 3) were less efficient for this reaction. When the reaction was carried out using iridium catalysts such as [Ir(cod)Cl]$_2$, we did not observe any C–H alkylation products (Table 1, entry 4). Further control experiments confirmed that the coupling process did not occur in the absence of the Rh catalyst (Table 1, entry 5) and the conversion was poor without NaHCO$_3$ (Table 1, entry 6). Other bases such as Na$_2$CO$_3$ led to dramatic erosions in yield (Table 1, entry 7). The non-polar solvent was required for the reaction and the use of xylene as a solvent led to a slightly lower yield (Table 1, entry 8). Lowering the temperature to 130 °C had a little impact on the reaction outcome (Table 1, entry 9); however, at a lower

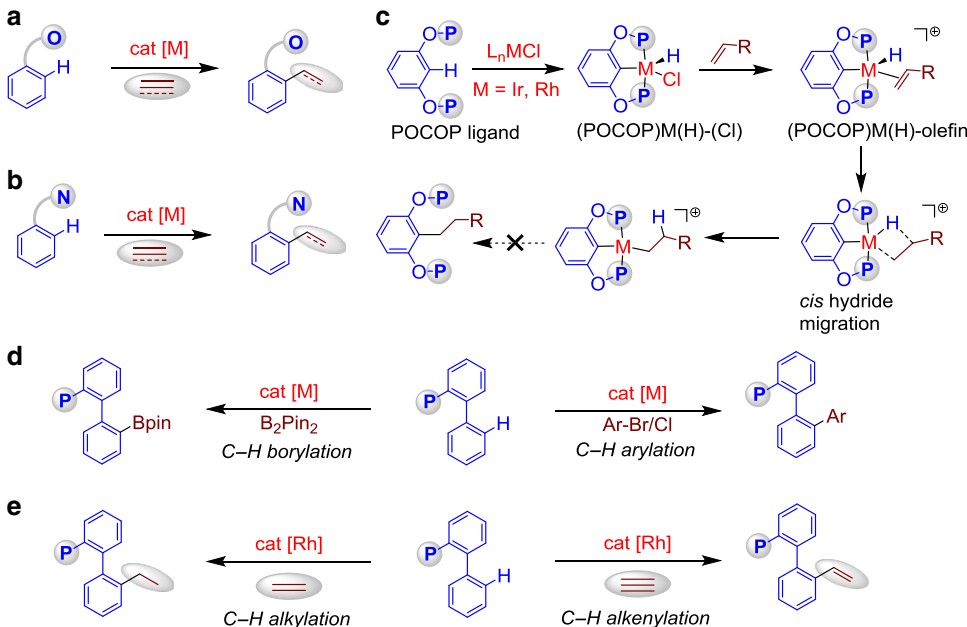

**Fig. 1** Towards a catalytic process for hydroarylation of alkenes and alkynes with diarylphosphines through P(III)-chelation-assisted C−H activation. **a** O-Chelation-assisted hydroarylation. **b** N-Chelation-assisted hydroarylation. **c** Precedent studies on P-chelated transition metal-olefin complexes. **d** Transition-metal-catalysed C−H borylation and arylation of diarylphosphines. **e** Rh-catalysed P(III)-chelation-assisted hydroarylation of alkenes and alkynes with diarylphosphines

**Table 1 Reaction optimization**

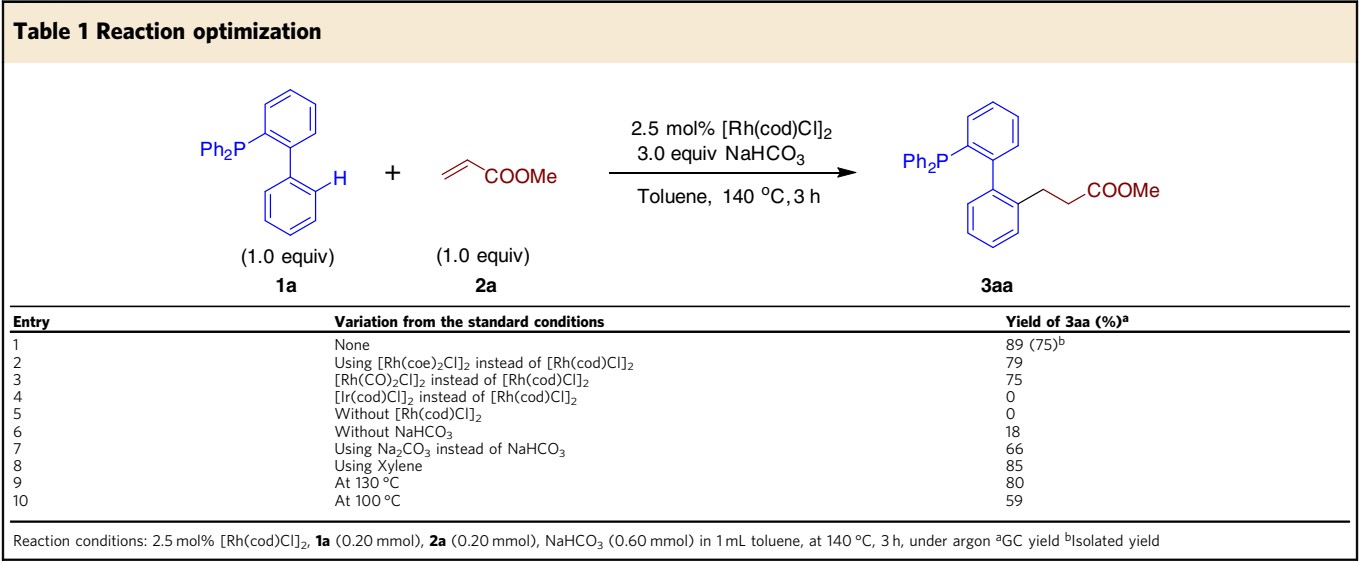

| Entry | Variation from the standard conditions | Yield of 3aa (%)[a] |
|---|---|---|
| 1 | None | 89 (75)[b] |
| 2 | Using [Rh(coe)$_2$Cl]$_2$ instead of [Rh(cod)Cl]$_2$ | 79 |
| 3 | [Rh(CO)$_2$Cl]$_2$ instead of [Rh(cod)Cl]$_2$ | 75 |
| 4 | [Ir(cod)Cl]$_2$ instead of [Rh(cod)Cl]$_2$ | 0 |
| 5 | Without [Rh(cod)Cl]$_2$ | 0 |
| 6 | Without NaHCO$_3$ | 18 |
| 7 | Using Na$_2$CO$_3$ instead of NaHCO$_3$ | 66 |
| 8 | Using Xylene | 85 |
| 9 | At 130 °C | 80 |
| 10 | At 100 °C | 59 |

Reaction conditions: 2.5 mol% [Rh(cod)Cl]$_2$, **1a** (0.20 mmol), **2a** (0.20 mmol), NaHCO$_3$ (0.60 mmol) in 1 mL toluene, at 140 °C, 3 h, under argon [a]GC yield [b]Isolated yield

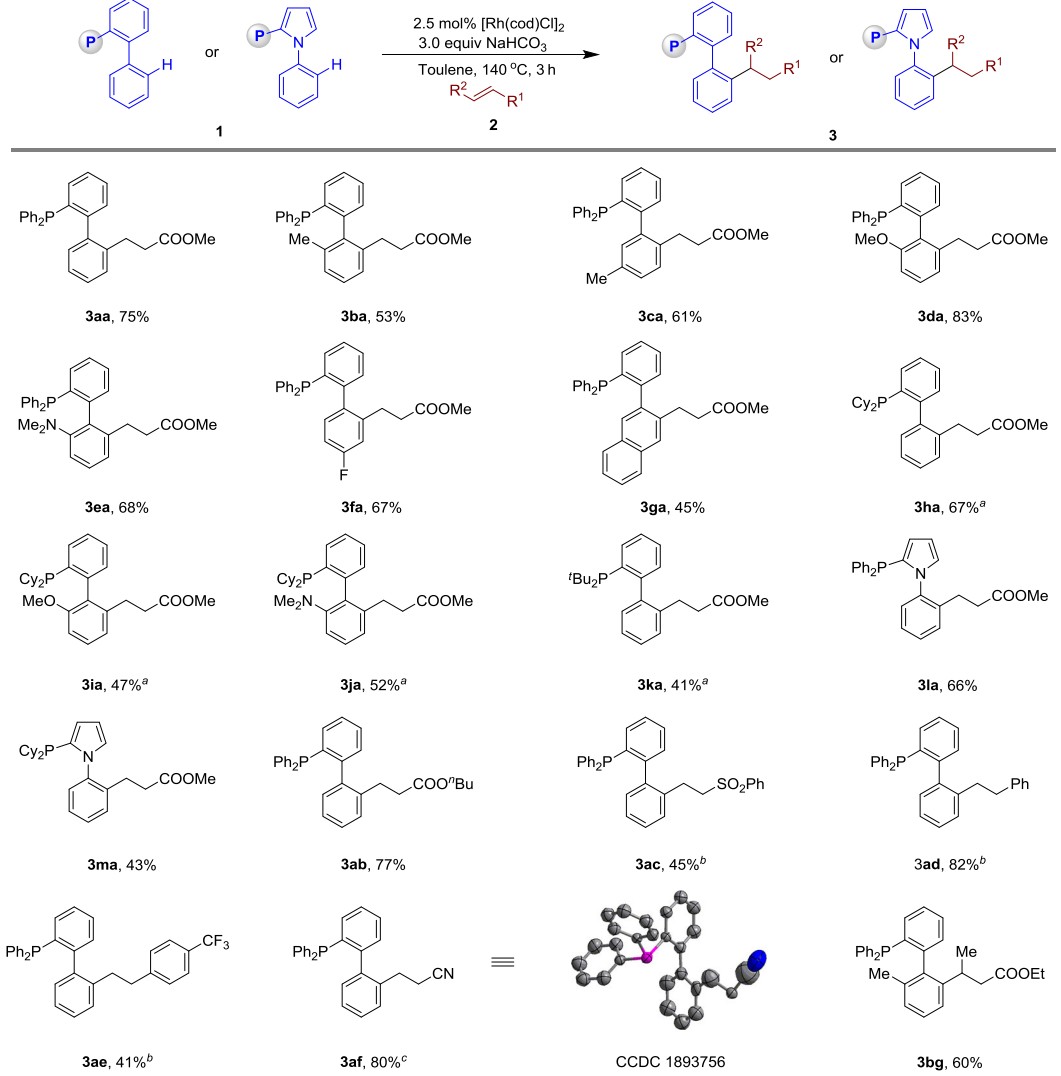

**Fig. 2** Rhodium-catalysed direct hydroarylation of alkenes with phosphines. Reaction conditions: 2.5 mol% [Rh(cod)Cl]$_2$, **1** (0.20 mmol), **2** (0.20 mmol), NaHCO$_3$ (0.60 mmol) in 1 mL toluene, at 140 °C, 3–4 h, under argon. All reported yields are isolated yields. [a]Using **2** (0.60 mmol) at 150 °C for 24 h. [b]5 mol% [Rh(coe)$_2$Cl]$_2$, **1** (0.20 mmol), **2** (0.60 mmol), K$_2$CO$_3$ (0.60 mmol) in 1 mL toluene, at 150 °C, 24 h, under argon. [c]Using 5 mol% [Rh(cod)Cl]$_2$, **2** (0.60 mmol) at 150 °C for 24 h

temperature, 100 °C the result was found much inferior than that observed under the optimal conditions (Table 1, entry 10).

**Scope of the methodology.** The scope of the mono-substituted alkylation products were examined using a wide range of phosphine ligands. As shown in Fig. 2, biaryl phosphines bearing Me (**1b**–**1c**), OMe (**1d**), NMe$_2$ (**1e**) and F (**1f**) substituents underwent facile hydroarylation with methyl acrylate (**2a**), affording the corresponding products **3ba**–**3fa** with 53–83% yields. Substrate **1g**, containing a naphthalen-2-yl group, was also compatible, and the C–H activation was located at the less sterically hindered C3 position. CyJohnPhos (**1h**), 2-(dicyclohexylphosphino)-2′-methoxybiphenyl (**1i**), and Davephos (**1j**) with a PCy$_2$ directing group produced hydroarylation products **3ha**–**3ja** with 47–67% yields. Furthermore, JohnPhos (**1k**), having a sterically hindered P$^t$Bu$_2$ group, produced the desired product **3ka** in modest yield. The

cataCxium ligand series, developed by Beller, such as cataCXium® PPh (**1l**) and cataCXium® PCy (**1m**), were compatible as well. Under the optimized conditions, phosphine **1a** was effectively coupled to olefins such as butyl acrylate (**2b**), (vinylsulfonyl)benzene (**2d**), and styrenes **2d–e** in high efficiency and regioselectivity. Reactions conducted with acrylonitrile (**2f**) formed a linear hydroarylation product **3af** with 80% yield, which was confirmed by X-ray analysis. Notably, internal olefins such as methyl (*E*)-but-2-enoate (**2g**) were tolerated as well.

Under slightly modified conditions, the reaction of phosphine **1a** (1.0 equiv) and 1-ethynyl-4-methoxybenzene (**4a**, 2.0 equiv) in the presence of 2.5 mol% [Rh(cod)Cl]$_2$ without NaHCO$_3$, at 120 °C under an Ar atmosphere in toluene, formed a Markovnikov hydroarylation adduct **5aa** in 85% yield (Fig. 3). Then we studied the scope of the hydroarylation reaction between phosphine ligands and alkynes for the synthesis of alkenyl-substituted products. Unlike the above reactivity, this olefination reaction is sensitive to the steric

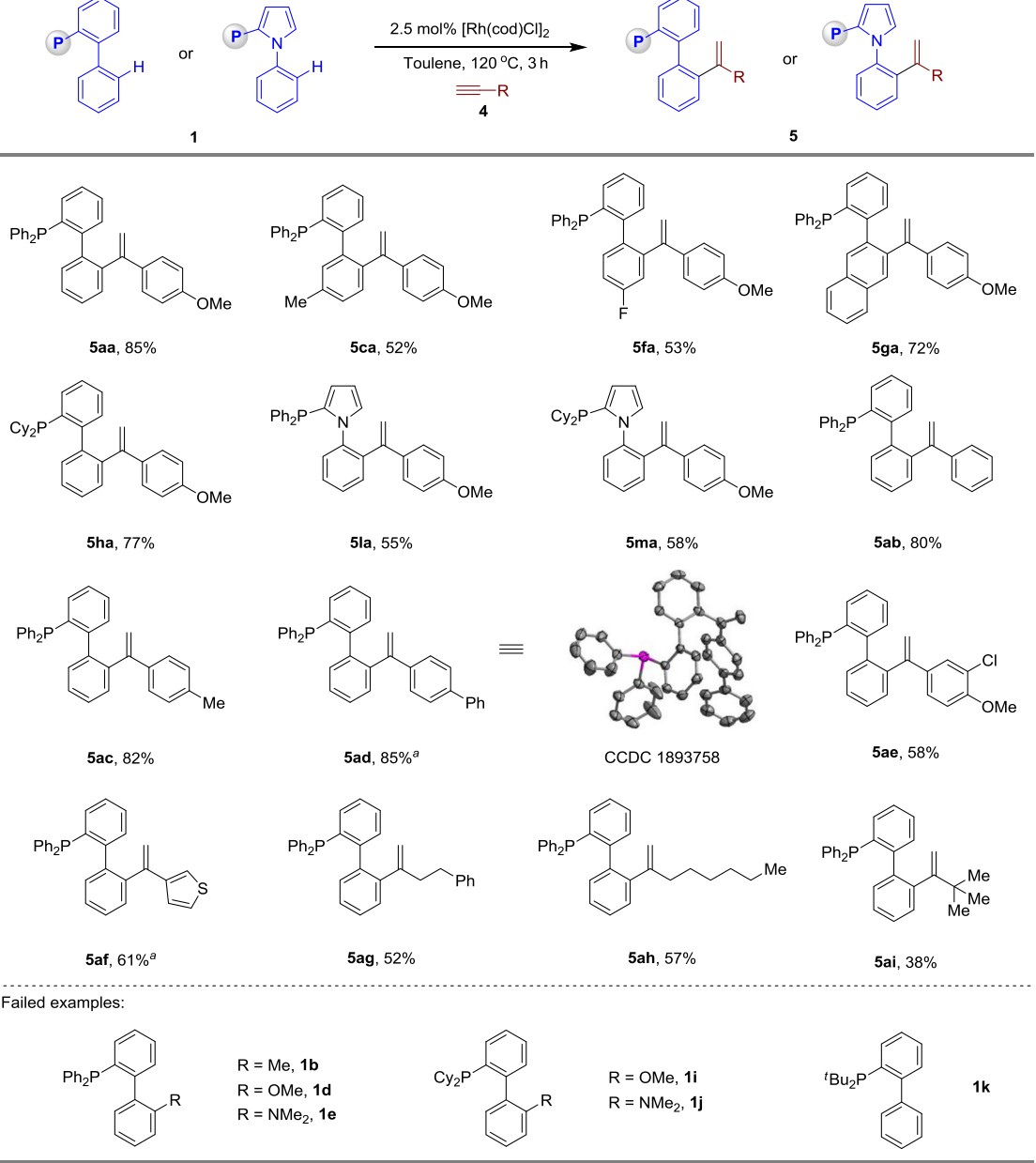

**Fig. 3** Rhodium-catalysed direct hydroarylation of alkynes with phosphines. Reaction conditions: 2.5 mol% [Rh(cod)Cl]$_2$, **1** (0.20 mmol), **4** (0.40 mmol) in 0.5 mL toluene, at 120 °C, 12 h, under argon. All reported yields are isolated yields. $^a$Using **1** (0.20 mmol), **4** (0.60 mmol)

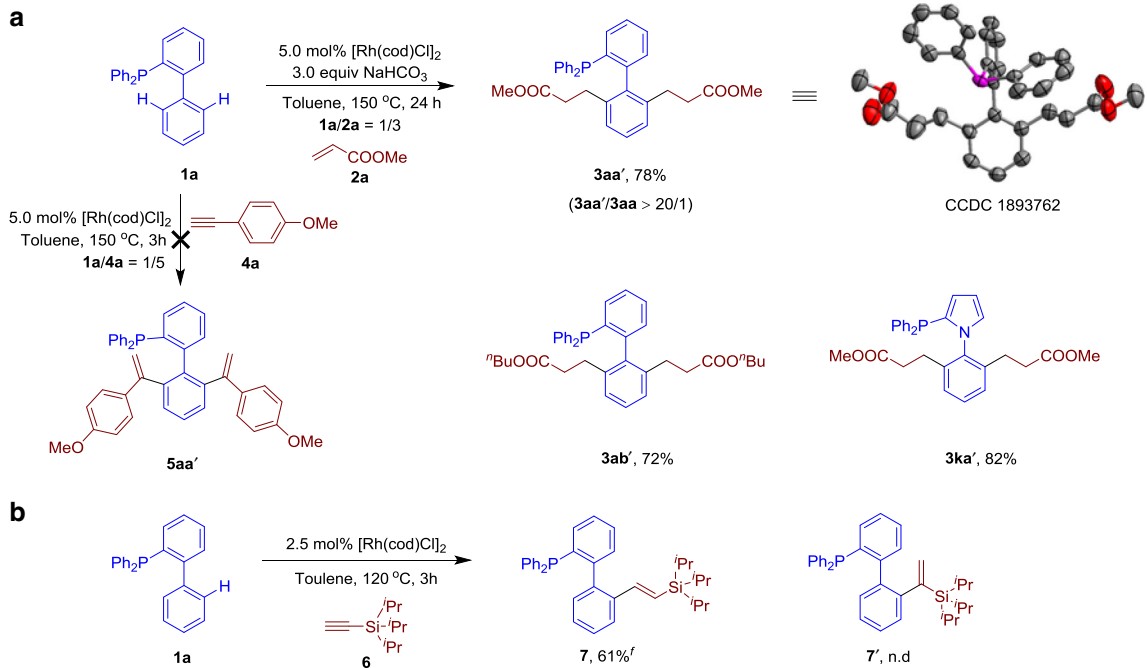

**Fig. 4** Further investigations. **a** Rh-catalysed di-selective hydroarylation of alkenes with phosphines. **b** Rh-catalysed *anti*-Markovnikov hydroxylation of alkyne **6** with phosphine **1a**

hindrance of the phosphines. Therefore, the C2-substituted phosphines **1b, 1d–e** and **1i–j** and the sterically hindered JohnPhos (**1k**) with the P*t*Bu₂ directing group were not tolerated. Phosphines such as **1c, 1f–h** and **1l–m** were applied in this C–H olefination reaction. Terminal alkynes, including phenylacetylene (**4b**), as well as derivatives that incorporate methyl (**4c**), phenyl (**4d**) and Cl (**4e**) on arenes, which were readily tolerated (**5ab–5ae**). As an example of a heteroaromatic alkyne, 3-ethynylthiophene (**4f**) was shown to be amenable to this protocol. Furthermore, the hydroarylation of alkyl-substituted terminal alkynes **4g–i** proceeded smoothly with branch-selectivity.

In addition to the mono-selective hydroarylation, we also investigated the disubstituted products. To increase the di-selectivity to a more synthetically useful level, phosphine **1a** was treated with 3.0 equiv of **2a** and 3.0 equiv of NaHCO₃ in toluene at 150 °C for 24 h to selectively form the disubstituted product **3aa′** with a 78% yield, which was confirmed by X-ray analysis (Fig. 4a). Other olefins such as butyl acrylate (**2b**) and other phosphines such as cataCXium® PCy (**1l**) produced the corresponding disubstituted hydroarylation products **3ab′** and **3ka′** in good yields. However, the C–H alkenylation reactions only displayed mono-selectivity. For instance, the reaction of phosphine **1a** with a large amount of alkyne **4a** still generated the mono-substituted product **5aa** and the corresponding disubstituted product **5aa′** was not detected. Besides the Markovnikov selectivity observed in Fig. 3, interestingly, an extremely bulky TIPS substituted terminal alkyne **6** only produced the product **7** in an *anti*-Markovnikov mode (Fig. 4b).

**Synthetic applications**. Besides diarylphosphines, this strategy can also be extended to binaphthyl-based chiral phosphine ligands. Notably, alkyl, ether and dialkylamino groups are widely used to adjust the steric structure in these phosphines. Based on this method, the alkyl groups could be rapidly and efficiently installed by P(III)-directed C–H alkylation of alkenes. For example, when (*R*)-H-MOP (**8**) was employed with olefin **2 h**, the

alkylation product **9** was generated in 56% yield without erosion of ee (Fig. 5a). To further modulate the steric properties of the substituent at alkyl substituents, olefins **2i** and **2j** with bulky substituents were employed, providing the compounds **10–11** with acceptable yields and excellent stereochemical reliability. To show the synthetic utility, we next tested them as ligands in transition metal-catalysed asymmetric reactions. As shown in Fig. 5b, following the procedures disclosed by the Hayashi research group on the Rh-catalysed arylation of related isatins **12a–b** with PhB(OH)₂ (**13**) using the best ligand (*R*)-MeO-MOP, alcohols **14a–b** were obtained in good yields with 75%[63] and 89% ee[64], respectively. Furthermore, when the modified ligands **9–11** were screened, the enantioselectivity of the product **12** could be dramatically improved to 84% ee. In addition, compound **9** showed a better reactivity for the synthesis of the product **12b**.

## Discussion

To probe the reaction mechanism, we conducted a series of experimental investigations (Fig. 6). Reactions of the [Rh(cod)Cl]₂ dimer with phosphine **1a** ($\delta_P = -13.6$ ppm) in toluene at room temperature generated a yellow complex **15** ($\delta_P = 20.5$ ppm), as confirmed by X-ray analysis and ³¹P NMR spectroscopy. Detected by the ³¹P NMR spectrum, no new signal appeared when the solution **6** was heated in D8-toluene at 120 °C[65,66]. Addition of alkene **2a** to the above system generated two new signals, confirmed as intermediate **16** ($\delta_P = 36.7$ ppm) and the product **3aa′** ($\delta_P = -15.0$ ppm). Following a similar protocol for alkyne **4a** resulted in the formation of compounds **17** ($\delta_P = 35.7$ ppm) and **5aa** ($\delta_P = -14.9$ ppm). Further detected by high-resolution mass spectrometry (HRMS), the analysed samples **16** and **17** showed diminished signals for the precursors of products **3aa′** and **5aa**. This result indicates that the C–H activation event is triggered by the addition of the alkene or alkyne, and phosphine chelation with the rhodium species is crucial for this regio-controlled hydroarylation.

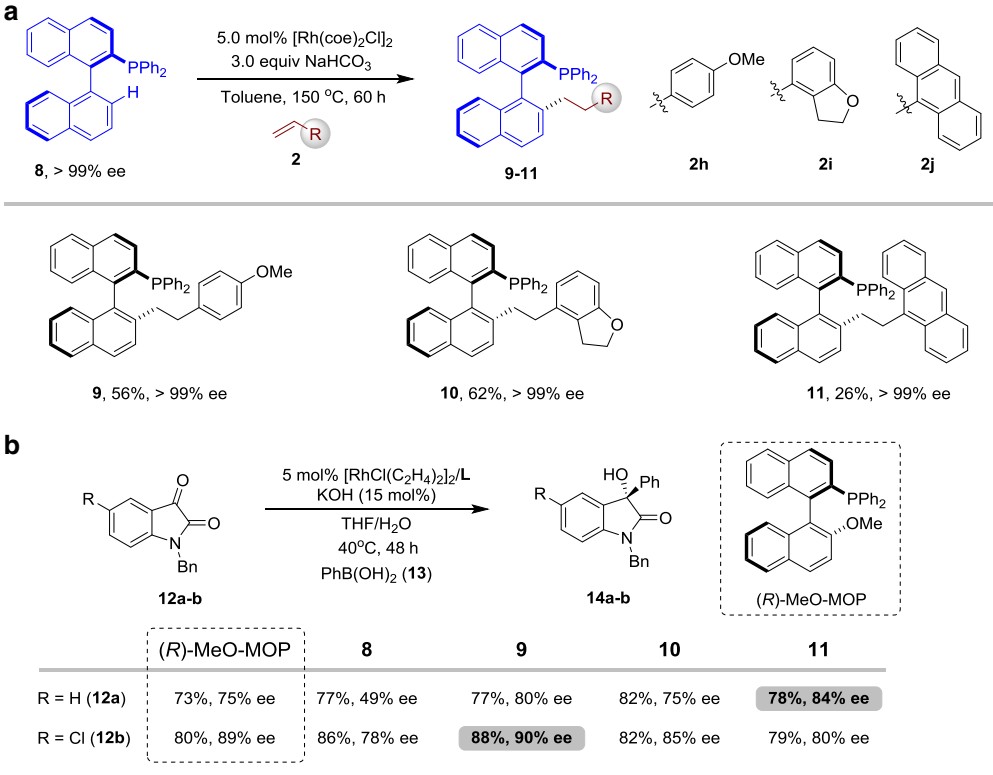

**Fig. 5** Testing the developed binaphthyl-based chiral phosphine ligands. **a** Rh-catalysed C–H alkylation of with (R)-H-MOP (**8**). **b** Rh-catalysed asymmetric addition of arylboronic acids to isatins

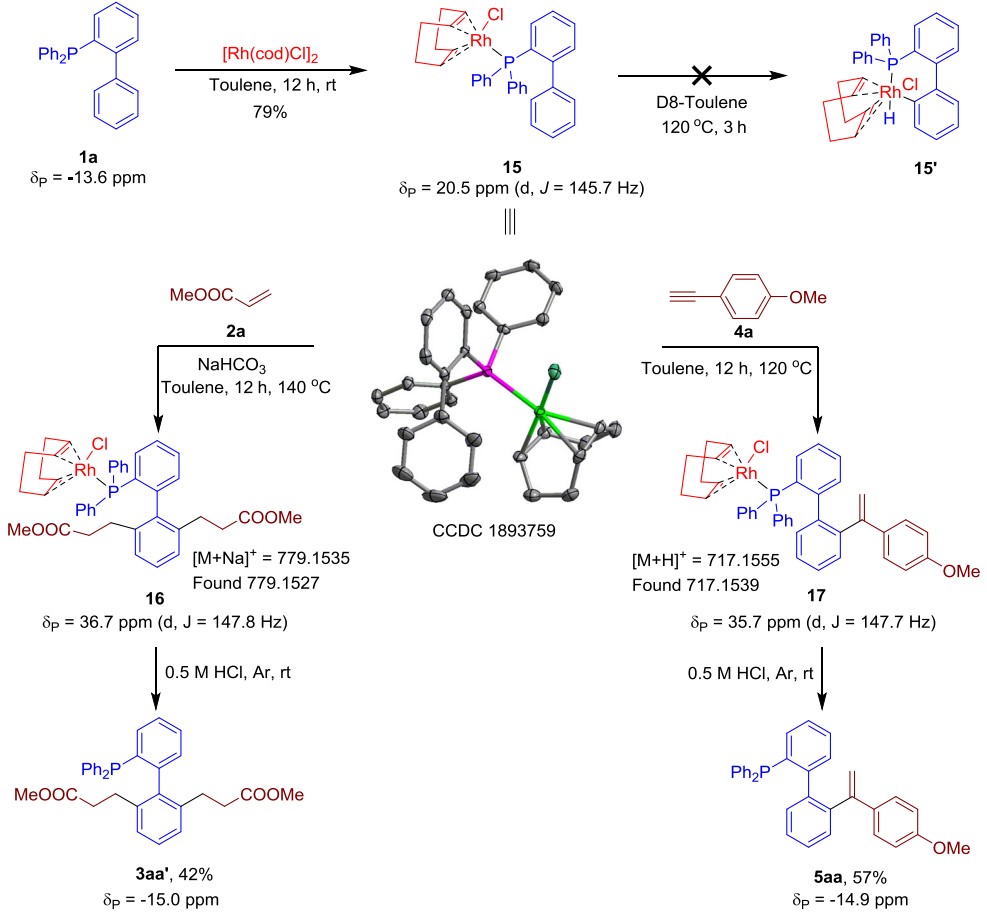

**Fig. 6** Mechanistic studies experiments. Investigation of the reaction intermediates by X-ray analysis, $^{31}$P NMR spectroscopy and HRMS

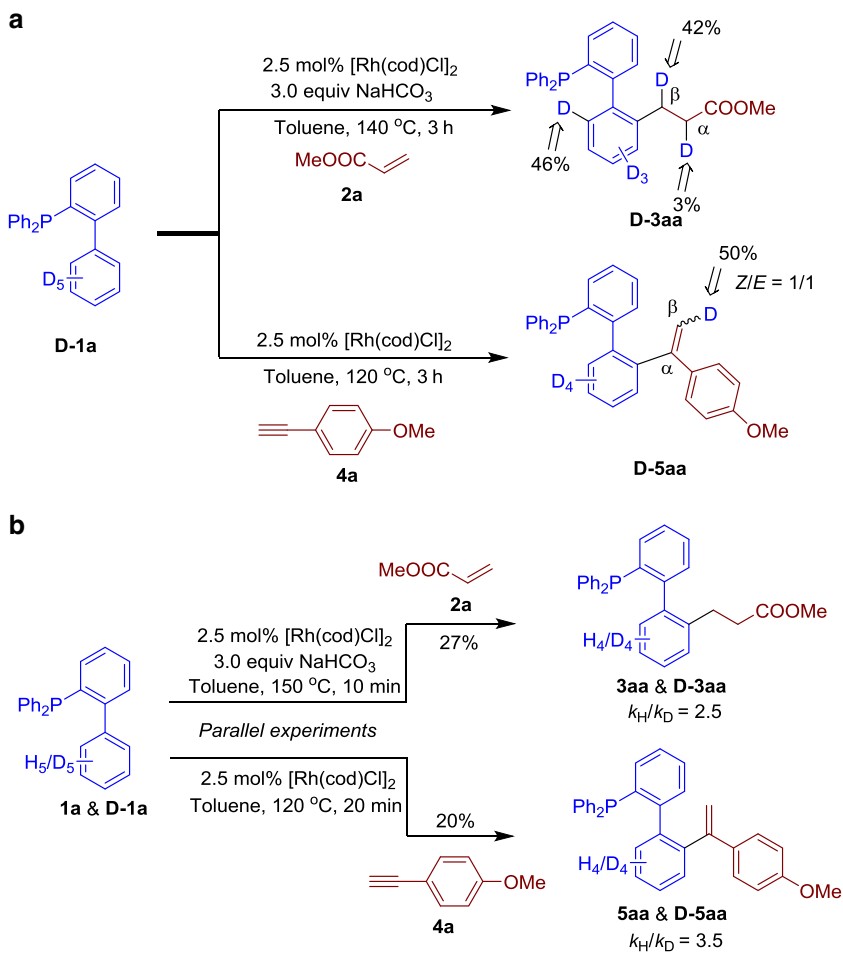

**Fig. 7** Deuterium labelling experiments. **a** Deuteration experiments of D-1a with alkene 2a and alkyne 4a. **b** The kinetic isotope effects of C–H alkylation and alkenylation

To gain further insight into the reaction mechanism, deuterated experiments were carried out by **D-1a** with **2a** in the standard reaction conditions. [1]H NMR analysis revealed that D/H-exchange was detected on the D5-benzonic core of **1a**, and 42% and 3% deuterium incorporation at the β and α positions, respectively, were observed after 3 h. Moreover, 50% deuterium incorporation was detected at the β-olefinic position of **D-5aa**, without stereoselectivity (Fig. 7a). In addition, the KIE values of the C–H activation in hydroarylation of **2a** and **4a** with phosphine **1a** were 2.5 and 3.5, respectively, revealing that the C–H cleavage is slow, and involved as a rate-determining step in both alkylation and alkenylation process (Fig. 7b).

Based upon the results herein and the precedent reports, plausible reaction pathways are shown in Fig. 8. The rhodium species RhX first coordinates to the P atom of phosphine **D5–1**, which leads to the formation of complex **A**. A reversible coordination with alkene **2** or alkyne **4** delivers the intermediate **B**[45]. In the case of methyl acrylate (**2a**), subsequent C–H activation takes place and delivers the alkene-hydride complex **C**. The deuterium labelling experiments demonstrates the *cis* insertion of the Rh–D bond into **2a**, which first occurs at the α-position to generate intermediate **D**[46], then undergoes reversible β-hydride elimination to form the olefin complex **E**[67] and eventually produces the β-addition complex **F**. In the alkyne **4a** system, the Rh–alkyne complex **B** undergoes C–H activation to form rhodacycle **C′**, which undergoes a hydride addition at the α-position to generate intermediate **D′**. If reductive elimination happens at

this step, we should only observe the *trans*-hydroarylation product **D-5aa**. The nonstereospecific deuteration indicates that the species **F′** is possibly generated from carbenoid **E′**[68] via isomerization of the **D′** species. Then, intermediates **F** and **F′** produce complex **G** via reductive elimination, which generates the desired products **3** and **5** and then reforms catalytic species **A** via exchange with another phosphine molecule.

In summary, we developed an effective rhodium-catalysed system that can activate the aromatic C–H bonds of phosphines with alkenes and alkynes through a hydroarylation process. A diverse class of alkyl and phosphine ligands with different steric and electronic properties are obtained in moderate to good yields with excellent site-selectivity. Additional applications of the developed ligand libraries and more detailed mechanistic studies are also ongoing in our laboratory.

## Methods

**General procedures for synthesis of 3**. In an oven-dried Schlenk tube, **1** (1.0 equiv, 0.20 mmol), **2** (1.0 equiv, 0.20 mmol), [Rh(cod)Cl]₂ (2.5 mol%, 2.5 mg, 0.005 mmol), NaHCO₃(3.0 equiv, 50.4 mg, 0.60 mmol) were dissolved in freshly distilled toluene (1.0 mL). The mixture was stirred at 140 °C under argon for 3 h. Upon the completion of the reaction, the solvent was removed. The crude mixture was directly subjected to column chromatography on silica gel using petrol ether/EtOAc as eluent to give the desired products.

**General procedures for synthesis of 5**. In an oven-dried Schlenk tube, **1** (1.0 equiv, 0.20 mmol), **4** (2.0 equiv, 0.40 mmol), [Rh(cod)Cl]₂ (2.5 mol%, 2.5 mg, 0.005 mmol) were dissolved in freshly distilled toluene (1.0 mL). The mixture was

**Fig. 8** Proposed mechanism. A tentative reaction mechanism involves coordination of Rh species with P atom, C–H activation, insertion of alkenes and alkynes to form the desired products

stirred at 120 °C under argon for 12 h. Upon the completion of the reaction, the solvent was removed. The crude mixture was directly subjected to column chromatography on silica gel using petrol ether/EtOAc as eluent to give the desired products.

## Data availability

The authors declare that the data supporting the findings of this study are available within the article and Supplementary Information file, or from the corresponding author upon reasonable request. The X-ray crystallographic coordinates for structures reported in this study have been deposited at the Cambridge Crystallographic Data Centre (CCDC), under deposition numbers CCDC 1893756, CCDC 1893758, CCDC 1893759 and CCDC 1893762. These data can be obtained free of charge from The Cambridge Crystallographic Data Centre via www.ccdc.cam.ac.uk/data_request/cif.

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

## Acknowledgements

We thank the "1000-Youth Talents Plan", the National Natural Science Foundation of China (Grants 21672097) and the "Innovation & Entrepreneurship Talents Plan" of Jiangsu Province for their financial support.

## Author contributions

D.W., B.D., Y.W., J.Q. and J.Z. performed the experiments. Y.Z. performed the crystallographic studies. Z.S. conceived the concept, directed the project and wrote the paper.

## Additional information

**Competing interests:** The authors declare no competing interests.

