## [Peer Review File · Nature Communications]

Reviewers' comments:

Reviewer #1 (Remarks to the Author):

This is a potentially quite useful manuscript from Shi and co-workers which describes a valuable method for the elaboration of the biarylphosphine ligand scaffold. Given the broad utility of this scaffold in a range of reactions, approaches that can functionalize this after the phosphine have been introduced could be potentially very powerful.

There is a reasonable amount of precedent in the literature for the transformation ie the same group have shown direct arylation of similar scaffolds, also using rhodium catalysis. In addition, several borylation reactions have been reported which can functionalize at the same position. However, this manuscript covers both alkylation and alkenylation, depending on whether alkenes or alkynes are used. The conditions are simple and the scope is fairly broad. It is useful that they show protocols to achieve both mono alkylation as well as dialkylation. The mechanistic studies all seem well executed and reasonable and the proposed catalytic cycle certainly seems likely given the deuterium experiments that were performed.

Given this broad scope and the potential utility in ligand evaluation, I believe this is suitable for publication in nature communications after addressing the following questions/comments.

Comments I have that I think would improve the manuscript, particularly since the usefulness of this is its application for practical ligand synthesis, rather than any particular conceptual novelty:

1. In the intro scheme it would be useful to have some scheme showing the previous work that has been done on direct C-H activation on the diarylphosphine scaffold. These are mentioned in the text but I think given the relevance to the current work it would be useful to have them in the scheme also.
2. What is the outcome if a meta-substituent is on the bottom aryl ring, do you get mixtures of regioisomers? Does this depend on what the substituent is? I think it would be very useful to show a couple of examples at least of these substrates.
3. Is it possible to have any kind of substitution on the Michael acceptor, like a methyl in either the a or b positions?

Reviewer #2 (Remarks to the Author):

Shi and co-workers report the phosphine-directed hydroarylation of alkenes and alkynes. The reaction works for a range of biaryl phosphines with electron deficient alkenes and a range of terminal alkynes. The authors conclude with some studies that provide insights into a proposed mechanism.

This manuscript is well-written and the conclusions are well-justified and this manuscript should be published after minor revisions. The suggested changes are below:

1) The numbering system in the text is difficult to follow and remember as one proceeds to a later part of the manuscript. I think this would be much clearer and easier to follow along if the authors provided an additional figure that showed the substrate numbering system using R's on the P atoms and any other places with substituents. As is, the text is complex to sort through. This is especially true when the authors are saying which substrates do not work since you cannot look at a drawn product.

2) In figure 3, the biaryl representation of 1 at the top is not general enough to include the Catacium class of phosphines. There should be two options shown for the biphenyl and for the catacium.

3) On the top of page 5, the authors refer to 5ma. I cannot find this product in the table. Is the number correct? Is it 5na?

4) Figure 5: From reading the paper, it is not obvious if the formation of 11 and 12 happen at RT or elevated temperatures. This should be added. I found it in the SI, but should be here as well. Same for release of the Rh. What are conditions for this? It was not in the SI. It should be added there as well.

5) Top of Figure 5: Why did the authors use 120 C when most of the reactions are run at 140 C, and even the olefin hydroarylation in the same figure is done at 140 C.

6) Line 169 of the same page: tran should be trans.

7) In figure 6: The authors say they run the KIE experiments in parallel. There are a number of ways I can envision this meaning. Can you please add some clarification to the SI so it is clear how you run these experiments in parallel.

Supporting Information

8) Please add the solvent conditions for each compounds column chromatography so the reactions are readily reproduced.

9) It is not clear how the authors analyzed the deuterium labeling experiments and the KIE's. If you are relying on the integrations to determine the ratio of deuterio and non-deuterio products, long relaxation delays must be used (usually 10s, if this was done, please add this information to the SI).

It would also be nice to have a description of the approach for analysis including an expansion of the key peaks that shows the integrations, assigns those peaks and shows how the % deuterio and proteo are determined.

Reviewer #3 (Remarks to the Author):

The authors report a rhodium catalyzed C-H functionalization followed by addition to alkenes and alkynes. A fairly detailed mechanistic study is undertaken that includes some labeling experiments.

The work is well executed and the results are interesting, but I would not say they are particularly surprising at this point after so much work in this field over the past two decades.

This study examines a directed activation, now known with many directing groups and many metal catalysts. C-H activation on adjacent aryl groups is also well established. The products are interesting but I do not see a lot of novelty or unrealized capacity compared to earlier work.

The work definitely merits publication but I am not really seeing how this reaches the bar for Nature Communications.

Reviewer #1:

This is a potentially quite useful manuscript from Shi and co-workers which describes a valuable method for the elaboration of the biarylphosphine ligand scaffold. Given the broad utility of this scaffold in a range of reactions, approaches that can functionalize this after the phosphine have been introduced could be potentially very powerful.

There is a reasonable amount of precedent in the literature for the transformation ie the same group have shown direct arylation of similar scaffolds, also using rhodium catalysis. In addition, several borylation reactions have been reported which can functionalize at the same position. However, this manuscript covers both alkylation and alkenylation, depending on whether alkenes or alkynes are used. The conditions are simple and the scope is fairly broad. It is useful that they show protocols to achieve both mono alkylation as well as dialkylation. The mechanistic studies all seem well executed and reasonable and the proposed catalytic cycle certainly seems likely given the deuterium experiments that were performed.

Given this broad scope and the potential utility in ligand evaluation, I believe this is suitable for publication in nature communications after addressing the following questions/comments.

Answer: Thanks for such positive comments.

Comments I have that I think would improve the manuscript, particularly since the usefulness of this is its application for practical ligand synthesis, rather than any particular conceptual novelty:

1. In the intro scheme it would be useful to have some scheme showing the previous work that has been done on direct C-H activation on the diarylphosphine scaffold. These are mentioned in the text but I think given the relevance to the current work it would be useful to have them in the scheme also.

Answer: According to your suggestion, we added the direct C-H activation on the diarylphosphine scaffolds in Fig 1d.

2. What is the outcome if a meta-substituent is on the bottom aryl ring, do you get mixtures of regioisomers? Does this depend on what the substituent is? I think it would be very useful to show a couple of examples at least of these substrates.

Answer: Thanks for this suggestion! We added a biarylphosphine **1c** with a meta-methyl for C-H alkylation and alkenylation. We only observed the products **3ca** and **5ca**, in which the reaction occurred at the less sterically hindered C-H bond.

3. Is it possible to have any kind of substitution on the Michael acceptor, like a methyl in either the a or b positions?

Answer: Thanks for this kind suggestion! You may not notice that we showed such examples like **3bg** in Fig. 2.

Reviewer #2:

Shi and co-workers report the phosphine-directed hydroarylation of alkenes and alkynes. The reaction works for a range of biaryl phosphines with electron deficient alkenes and a range of terminal alkynes. The authors conclude with some studies that provide insights into a proposed mechanism.

This manuscript is well-written and the conclusions are well-justified and this manuscript should be published after minor revisions. The suggested changes are below:

Answer: Thanks for such positive comments.

1. The numbering system in the text is difficult to follow and remember as one proceeds to a later part of the manuscript. I think this would be much clearer and easier to follow along if the authors provided an additional figure that showed the substrate numbering system using R's on the P atoms and any other places with substituents. As is, the text is complex to sort through. This is especially true when the authors are saying which substrates do not work since you cannot look at a drawn product.

Answer: Thanks for this suggestion! This olefination reaction is sensitive to the steric hindrance of the phosphines. To make these failed substrates more clear, we drew them in Fig 3. Now, it seems that it's no necessary to draw an additional figure to show all substrates now.

2. In figure 3, the biarylrepresentation of 1 at the top is not general enough to include the Catacium class of phosphines. There should be two options shown for the biphenyl and for the cataxium.

Answer: According to your suggestions, we changed figures 2 & 3.

3. On the top of page 5, the authors refer to 5ma. I cannot find this product in the table. Is the number correct? Is it 5na?

Answer: We changed it.

4. Figure 5: From reading the paper, it is not obvious if the formation of 11 and 12 happen at RT or elevated temperatures. This should be added. I found it in the SI, but should be here as well. Same for release of the Rh. What are conditions for this? It was not in the SI. It should be added

there as well.

Answer: According to your suggestions, we redrew the Figure and made the reaction conditions more clear.

5. Top of Figure 5: Why did the authors use 120 C when most of the reactions are run at 140 C, and even the olefin hydroarylation in the same figure is done at 140 C.

Answer: Thanks for this question. Our reactions in Figure 5 were detected by in situ variable-temperature NMR spectroscopy, which can't reach 140 oC. When the reactions were run at 140 °C and then we took the mixtures for NMR, there is no obvious difference compared to 120 °C.

6. Line 169 of the same page: tran should be trans.

Answer: We changed it.

7. In figure 6: The authors say they run the KIE experiments in parallel. There are a number of ways I can envision this meaning. Can you please add some clarification to the SI so it is clear how you run these experiments in parallel.

Answer: We added the detailed procedures in SI. Thanks.

8. Please add the solvent conditions for each compounds column chromatography so the reactions are readily reproduced.

Answer: We added the detailed solvent conditions for each compounds column chromatography in SI. Thanks.

9. It is not clear how the authors analyzed the deuterium labeling experiments and the KIE's. If you are relying on the integrations to determine the ratio of deuterio and non-deuterio products, long relaxation delays must be used (usually 10s, if this was done, please add this information to the SI).

It would also be nice to have a description of the approach for analysis including an expansion of the key peaks that shows the integrations, assigns those peaks and shows how the % deuterio and proteo are determined.

Answer: According to your suggestions, the relevant data for the deuterium labeling experiments and the KIE's have been added in the supporting information.

Reviewer #3 (Remarks to the Author):

The authors report a rhodium catalyzed C-H functionalization followed by addition to alkenes

and alkynes. A fairly detailed mechanistic study is undertaken that includes some labeling experiments.

The work is well executed and the results are interesting, but I would not say they are particularly surprising at this point after so much work in this field over the past two decades. This study examines a directed activation, now known with many directing groups and many metal catalysts. C-H activation on adjacent aryl groups is also well established. The products are interesting but I do not see a lot of novelty or unrealized capacity compared to earlier work.

The work definitely merits publication but I am not really seeing how this reaches the bar for Nature Communications.

Answer: Thanks for your affirmation of the execution and results for our work. You can see the first and second referees gave us quite positive assessments. Your suggestions were very helpful for us to improve the quality of this paper. You raised concerns about the novelty and utility of this strategy. I would like to demonstrate them in details.

As you mentioned, this study examines a directed activation, now known with many directing groups and many metal catalysts. In 1993, Murai et al. reported a significant, pioneering work on the Ru-catalysed hydroarylation of alkenes with aromatic ketones by carbonyl-directed C-H activation. During the past 25 years, chelation assisted hydroarylation have been reported by many research groups. However, these reactions are only limited to N and O coordination functional groups. The control of the regioselectivity by other heteroatoms, especially P atom remains a challenge. In this paper, we showed the first example on the hydroarylation of alkenes and alkynes with tertiary phosphines through P(III)-chelation assisted C-H activation. A series of ligand libraries containing alkyl and alkenyl substituted groups with different steric and electronic properties were obtained in high yields.

Ligand modification is a powerful approach for designing new ligands with excellent reactivity. Based on your comments, we added a Figure to show the utility of this strategy during the revision. We ignored this important part in the former submission. You can see Figure 5, based on the outstanding properties of the parent scaffolds, the alkyl-substituted phosphines have been found to be powerful in Rh-catalyzed asymmetric reactions to improve the enantioselectivity rapidly. Our developed ligands 9 and 11 have been the best ones in rhodium-catalyzed asymmetric addition of arylboronic acids to isatins so far.

Therefore, I strongly hope you can reconsider the novelty and utility of our strategy in light of these points.

REVIEWERS' COMMENTS:

Reviewer #1 (Remarks to the Author):

The requested changes have now been made so I am satisfied with the manuscript. The only point to make is that there is a typo in the text - for the newly added scheme in the intro it is referred to in the text as 1d but is actually 1e.

REVIEWERS' COMMENTS:

Reviewer #1 (Remarks to the Author):

The requested changes have now been made so I am satisfied with the manuscript.

The only point to make is that there is a typo in the text - for the newly added scheme in the intro it is referred to in the text as 1d but is actually 1e.

Answer: Thanks for this suggestion! We changed it.